# One Health Approach: An Overview of Q Fever in Livestock, Wildlife and Humans in Asturias (Northwestern Spain)

**DOI:** 10.3390/ani11051395

**Published:** 2021-05-13

**Authors:** Alberto Espí, Ana del Cerro, Álvaro Oleaga, Mercedes Rodríguez-Pérez, Ceferino M. López, Ana Hurtado, Luís D. Rodríguez-Martínez, Jesús F. Barandika, Ana L. García-Pérez

**Affiliations:** 1Department of Animal Health, Regional Service for Agrofood Research and Development (SERIDA), 33394 Gijón, Spain; anadc@serida.org; 2Translational Microbiology Consolidated Group, Instituto de Investigación Sanitaria del Principado de Asturias (Health Research Institute of Asturias, ISPA), Av. del Hospital Universitario, s/n, 33011 Oviedo, Spain; alvaroleaga@yahoo.es (Á.O.); Mercedes.rodriguezp@sespa.es (M.R.-P.); 3SERPA—Sociedad de Servicios del Principado de Asturias S.A., 33202 Gijón, Spain; 4Department of Microbiology, Central Hospital of Asturias (HUCA), 33011 Oviedo, Spain; 5Department of Animal Pathology, Animal Health, Veterinary Faculty, University of Santiago de Compostela, 27071 Lugo, Spain; c.lopez@usc.es; 6Department of Animal Health, NEIKER-Basque Institute for Agricultural Research and Development, Basque Research and Technology Alliance (BRTA), 48160 Derio, Spain; ahurtado@neiker.eus (A.H.); jbarandika@neiker.eus (J.F.B.); agarcia@neiker.eus (A.L.G.-P.); 7Animal Health Laboratory (LSAPA), Government of Principado de Asturias, 33201 Gijón, Spain; LUISDARIO.RODRIGUEZMARTINEZ@asturias.org

**Keywords:** Q fever, *Coxiella burnetii*, seroprevalence, ruminants, wildlife, humans, dust, aerosols

## Abstract

**Simple Summary:**

We studied Q fever in an area of Spain where a significant number of human cases are diagnosed every year. Although animals are the only source of infection for people, this is the first study carried out in the autonomous community of Asturias that addresses in an integrated way the infection in domestic animals, wildlife and the environment as well as people. Our results revealed that a remarkable percentage of domestic ruminants and wild ungulates from all geographic areas of the region had been in contact with the infection’s causative agent (*Coxiella burnetii*). In addition, the bacteria could be detected in the air and/or the dust of livestock farms. Finally, a statistical analysis was carried out to investigate the risk factors (age, sex, geographical area, etc.) for the human population of the region. These findings will help local health authorities to focus on the origin of the problem and facilitate applying preventive measures in the affected livestock farms.

**Abstract:**

This study aimed to investigate the seroprevalence of *C. burnetii* in domestic ruminants, wild ungulates, as well as the current situation of Q fever in humans in a small region in northwestern Spain where a close contact at the wildlife–livestock–human interface exists, and information on *C. burnetii* infection is scarce. Seroprevalence of *C. burnetii* was 8.4% in sheep, 18.4% in cattle, and 24.4% in goats. Real-time PCR analysis of environmental samples collected in 25 livestock farms detected *Coxiella* DNA in dust and/or aerosols collected in 20 of them. Analysis of sera from 327 wild ungulates revealed lower seroprevalence than that found in domestic ruminants, with 8.4% of Iberian red deer, 7.3% chamois, 6.9% fallow deer, 5.5% European wild boar and 3.5% of roe deer harboring antibodies to *C. burnetii*. Exposure to the pathogen in humans was determined by IFAT analysis of 1312 blood samples collected from patients admitted at healthcare centers with Q fever compatible symptoms, such as fever and/or pneumonia. Results showed that 15.9% of the patients had IFAT titers ≥ 1/128 suggestive of probable acute infection. This study is an example of a One Health approach with medical and veterinary institutions involved in investigating zoonotic diseases.

## 1. Introduction

Q fever is a worldwide distributed zoonosis caused by *Coxiella burnetii*, a small intracellular bacterium belonging to γ-Proteobacteria [1,2] that infects a wide range of animal species, including mammals, birds and arthropods. People are infected through inhalation of aerosols contaminated with the bacteria expelled by infected animals during abortion or normal deliveries. Among domestic ruminants, sheep and goats are considered the main reservoirs of the infection and the principal source of human outbreaks [3,4]. *C. burnetii* has been reported in over a hundred wild mammal species that can be reservoirs for livestock and humans [5]. Reported cases of Q fever linked to exposure to wildlife can be associated with changes in the wildlife–human interactions leading to an increased risk of interspecies transmission [6]. Ticks are not essential in the domestic cycle of *C. burnetii* infection in livestock but may play a significant role in the wild cycle of transmission of coxiellosis among wild vertebrates [1,3].

Human Q fever is a public health problem worldwide [1,7]. After the outbreak in the Netherlands (2007–2010), linked to goat farms and involving more than 4000 people [2,4], the efforts devoted to studying this zoonosis have increased significantly. In Spain, the disease is considered endemic in several regions [7,8]. A systematic review recently conducted [9] showed significant differences in disease manifestations according to the geographical location. In the northern areas of Spain, pneumonia was the predominant symptom, while in the central and southern areas, isolated fever followed by hepatitis was the most frequent clinical form. In Asturias (northern Spain), pneumonia is the main clinical presentation of Q fever [10,11,12], and a relatively high risk of exposure to *C. burnetii* in the population in Asturias has been reported [7,9,11]. In fact, the fatality rate associated with *C. burnetii* infection in the region in the period 1997–2015 (7.69 per 100) was the highest compared to other Spanish regions [7]. There are very few studies on the exposure of wildlife to *C. burnetii*, in which red deer is the only species investigated [13,14]. In addition, little is known about the role of domestic ruminants as reservoirs of *C. burnetii* in Asturias. When dealing with zoonotic diseases like Q fever, a coordinated approach involving human and animal health professionals working together from a unique perspective (One Health) is needed to reduce the risk of infection for both humans and animals. This approach should also consider the environmental risk associated with the domestic and wild cycle of Q fever, particularly in regions of high nature tourist value like Asturias, where the human population is in close contact with nature, and consequently, with livestock and wildlife.

This study aimed to investigate the exposure to *C. burnetii* in domestic ruminants, wild ungulates and humans in northwestern Spain from a One Health perspective through the work of a multidisciplinary team integrated by microbiologists, veterinarians and epidemiologists.

## 2. Materials and Methods

### 2.1. Study Area

The study was carried out in the principality of Asturias, an autonomous community of 10,604 km^2^ located in northwestern Spain with a population of 1,022,670 inhabitants [15]. The region can be divided into three different geographical areas: western, central and eastern Asturias, separated by large north-to-south oriented valleys running through the Cantabrian mountain range. The predominant climate is temperate oceanic [16], which favors developing deciduous and mixed forests interspersed with open pastures and meadows as the characteristic vegetation of this region. Livestock and wildlife are abundant in the region.

### 2.2. Animal and Human Population Investigated and Sample Collection

#### 2.2.1. Livestock

Livestock activity in Asturias has a long tradition and a significant impact on the economy. The last census recorded 392,789 cattle, 46,004 sheep and 31,023 goats [17]. Beef cattle have progressively increased their census (70% of the total) at the expense of dairy cattle (30%). The vast majority of sheep and goats are meat breeds, and flocks are widely dispersed in the region, with a total of 3705 sheep and 1221 goat herds [17] holding an average of 12 and 25 animals per farm respectively. Lambing/kidding season in sheep and goats concentrates in spring, though a few intensive dairy herds can have more than one lambing season per year. Parturitions in dairy cattle can occur along the year, whereas in beef cattle, they mainly concentrate in spring and early summer but can also occur in other seasons.

The sample size was calculated to estimate the prevalence of an infection with a 95% confidence level, for an expected prevalence of 10%, an absolute error of 5% and a normal population distribution. This required 139 bovine samples, 138 for sheep and 138 for goats. Ruminant blood samples were obtained from the jugular vein in sheep and goats and from the medial coccygeal vein in cattle. Blood was collected in plain tubes without anticoagulant by the veterinarians in charge of the Livestock Official Sanitary Campaigns and then submitted to the Animal Health Laboratory of the Principality of Asturias (LSAPA), and ca. 1% of them were selected by systematic random sampling. For sheep and goats due to the annual organization of Livestock Official Sanitary Campaigns in Asturias, sera from several years had to be compiled to reach the calculated sample size and achieve geographical representation of the different areas, as follows: 2016 (*n* = 60), 2017 (*n* =74) y 2018 (*n* = 20) for sheep, and 2015 (*n* = 44), 2016 (*n* = 52), 2017 (*n* =14) y 2018 (*n* = 25) for goats. For cattle, samples were all collected in 2018. Finally, samples of 154 sheep, 135 goats and 163 cows were subjected to serological analysis. All sera were collected from females older than 6 months for sheep and goats or older than 12 months in the case of cattle. Serum was obtained by centrifugation and stored at −20 °C until serological analysis.

Once the serological survey was completed, and with the aim of checking the presence of *C. burnetii* DNA in ruminant farms, 25 farmers, who did not know about the status of *C. burnetii* infection in their farms, agreed to participate voluntarily in the study. Seven dairy cattle farms, 5 goat farms, 2 sheep farms and 11 mixed flocks (with sheep, goats and/or cattle) were visited once between February and October 2019. In each farm, aerosols were taken inside the animal premises using the air sampler “MD8” Sartorius (Goettingen, Germany), performing an aspiration of 50 L/min air for 10 min. Dust samples were taken from different surfaces of animal premises with sterile swabs to detect the presence of *C. burnetii* DNA by real-time PCR.

#### 2.2.2. Wildlife

The percentage of protected areas in the principality of Asturias amounts to almost 22 percent of its territory, which in terms of surface area represents 228,879 hectares. These areas harbor a high percentage of the continental vertebrate species present in Spain (67%). Thus, faunistic richness in Asturias is high. A total of 327 blood samples from wild ungulates were included in the study (83 Iberian red deer, 57 roe deer, 41 Cantabrian chamois, 73 fallow deer and 73 European wild boars). Roe deer predominate in the west of the territory, fallow deer in the east, and the remaining species (red deer, chamois and wild boar) are present throughout the territory. Blood samples were collected in hunting seasons between 2004 and 2018 in the frame of SERIDA’s research projects related to wildlife populations. After blood centrifugation at the laboratory, sera were kept at–20 °C until serological analyses.

#### 2.2.3. Human Population Investigated

Blood samples were collected from patients who attended outpatient health services with compatible symptoms of Q fever (based on physicians’ criteria) to investigate the presence of antibodies against *C. burnetii*. A total of 1312 samples were submitted throughout 2018 to the Microbiology Service of the Central University Hospital of Asturias (HUCA) from 6 of the 8 Health Areas (HA) of the region (Occidente, Suroccidente, Oviedo, Mieres, Langreo, Oriente). Data collected included age, gender, HA and month of sampling.

### 2.3. Serological Analyses

#### 2.3.1. Animal Sera

An indirect ELISA test (PrioCHECK™ ruminant Q fever Ab plate kit, Thermo Fisher Scientific) was performed according to the manufacturer’s instructions. This commercial kit uses protein G as a conjugate, valid to analyze sera of wild ungulates, as reported elsewhere [18]. Antibody results in animal sera were expressed by titers based on the calculation of the sample/positive ratio (S/P = OD sample − ODm NC/ODm PC − ODm NC). Titers equal to or greater than 1:40 were considered positive.

#### 2.3.2. Human Sera

Sera from patients were first analyzed by indirect chemiluminescent immunoassay (CLIA) (*Coxiella burnetii* VirClia©, Vircell, Granada, Spain) to determine the presence of specific IgG antibodies against *C. burnetii* phase II in serum or plasma. Later, samples with a CLIA-positive result were titrated by indirect immunofluorescence assay (IFA) to detect antibodies anti-phase II (I + II IFA IgG/IgM/IgA, Vircell, Granada, Spain). Since a second sample was not taken 2–4 weeks apart to study seroconversion, a single IFA-positive convalescent serum IgG phase II ≥ 1:128 in a patient with compatible symptoms of Q fever of over 1 week’s duration was considered a probable acute infection [19].

#### 2.3.3. Molecular Analyses

Before DNA extraction, dust swabs were treated with 300 µL of TE buffer (Tris base 10 mM, EDTA 1 mM, pH 8) before being mixed with ATL and proteinase K for 1 h at 56 °C, and then, DNA extraction continued using the QIAmp DNA blood mini kit (Qiagen, Hilden, Germany). For aerosol samples, gelatine filters used in the air sampler device were treated with 2 mL of ATL buffer until gelatine was dissolved. This solution was mixed with 500 µL of buffer ATL and vortexed, centrifuged and heated at 56 °C. Then, two aliquots of 1 mL each were taken, and 50 µL of proteinase K (8 mg/mL) were added to each one, and the mixture was incubated for 1 h at 56 °C. The extraction process continued using QIAmp DNA blood mini kit (Qiagen, Hilden, Germany) following the manufacturer’s instructions. Negative extraction controls were included every 10 samples to rule out DNA contamination. The presence of *C. burnetii* DNA was investigated by a real-time PCR amplification targeting the transposon-like repetitive region IS1111 of the *C. burnetii* genome [20]. A commercial internal amplification control (IAC) (TaqMan^®®^ exogenous internal positive control, Thermo Fisher Scientific) was included in the assay to monitor for PCR inhibitors.

#### 2.3.4. Statistical Analysis

Logistic regression was used to analyze the possible influence of the different potential risk factors studied over seroprevalence against *C. burnetii* in domestic ruminants, i.e., animal species (categorical; sheep, goats, cattle), period of sampling (categorical; spring, summer, autumn/winter), year (categorical; year of sampling, applicable to sheep and goats only), herd size (categorical; <50 animals, >50 animals), and geographical location (categorical; east, central and western Asturias). Similarly, for wild ungulates, the animal species (categorical; red deer, roe deer, fallow deer, chamois, wild boar), the year of sampling (categorical; 2004–2007, 2008–2018) and the geographical area (categorical; east, west, all territory) were considered for the analysis.

In addition, the influence of risk factors over the presence of *C. burnetii* DNA in aerosols or dust was analyzed in 25 livestock farms using logistic regression. The variables included in the model were animal species (categorical; sheep, goats, cattle), production (categorical; milk, meat), herd size (categorical; <50 animals, >50 animals), geographical location (categorical; east, central and western Asturias), the month of sampling (categorical; February, March, April, May, July, August, September, October), recent abortions (categorical; yes, no), biosecurity measures implemented in the farm (categorical; poor, moderate, good) and livestock housing (categorical; old, modern, adapted shed). The final model was selected as the one with the lowest Akaike’s information criterion (AIC) value from all of the models performed. Odds ratio (OR) values were computed by raising “e” to the power of the logistic coefficient over the reference category. The seroprevalence against *C. burnetii* was calculated for each animal species. Statistical uncertainty was assessed by estimating the 95% confidence interval (CI) for each of the proportions according to the following formula: CI 95% = 1.96 [*p* (1 − *p*)/*n*]1/2, where *p* is the seroprevalence and *n* is the sample size.

To analyze human infection, patient data were grouped into categories, such as age (1–40, 41–60, 61–80, 81–100 years old), sampling period (March–May, June–October, November–February), gender (male/female), and geographical location (east Asturias (HA of Langreo and Oriente), central Asturias (HA of Oviedo and Mieres), and western Asturias (HA of Occidente and Suroccidente). A hierarchical cluster analysis was performed to determine the natural groupings of variables regarding seropositivity to *C. burnetii*. Homogeneous clusters of these categorical variables were identified using the ClustOfVar R package [21]. FactoMine R package for Multiple correspondence analysis (MCA) [22] was used to explore for associations between categories of qualitative variables related to demographic (age, gender), temporal (month of sampling), and geographical variables with Q fever acute infection (patients with symptoms compatible with Q fever and with IFA serology positive). MCA is an analytical method used to detect and display the underlying structure of a set of nominal categorical data using Euclidean distances. MCA graphically displays data relationships. Data are converted to a K-by-K table of all pairwise tabulations and represented on a two-dimensional graph where more proximal variables show a more similar distribution. The human census was compiled by geographical location, and the incidence of acute Q fever per 100,000 inhabitants was calculated. All statistical analyses were performed using the statistical software R version 4.0.20 [23].

#### 2.3.5. Ethics Statement

Anonymized animal and human data were provided by LSAPA and the Microbiology Service of HUCA, respectively. Blood sampling from domestic ruminants was carried out as part of the Livestock Official Sanitation Campaign, and human blood samples were taken in the course of disease diagnosis; therefore, written consent from farmers and patients was not required. The study protocol was approved by the Investigation Ethics Committee of the Principality of Asturias (Nº 274/19 for the animal study and 125/17 for the human study).

## 3. Results

### 3.1. Seroprevalence in Livestock

Overall, *C. burnetii* seroprevalence was 8.4% (95% CI: 5–14) in sheep, 18.4% (95% CI: 13–25) in cattle, and 24.4% (95% CI: 18–32) in goats (Table 1 and Appendix A). Geographically, seroprevalence in sheep was slightly higher in eastern Asturias (15.2%), whereas in goats, values were higher in eastern and central Asturias (29.1% and 28.0%, respectively). In general, higher prevalences were observed in areas where sheep and goat census are larger (Table 1). Conversely, the highest seroprevalence in cattle was found in western Asturias (24.4%), where the cattle census is slightly larger (Table 1).

Logistic regression models identified sampling year and flock size as variables associated with seropositivity in sheep (Table 2). Prevalence was significantly higher in 2018 than in other years (*p* = 0.0339; OR 7.48), and Q fever infection was associated with flocks with more than 50 animals (*p* = 0.0017; OR 7.18). In goats, the geographical location of the herd explained the prevalence (Table 2). Hence, flocks in the eastern region had a significantly higher prevalence than flocks in the western region (*p* = 0.0309; OR 9.65), and those in central Asturias marginally higher than herds in the western region (*p* = 0.0866; OR 6.70). No explanatory variables were found in cattle.

### 3.2. Investigation of C. burnetii DNA in Animal Premises

Twenty of the 25 farms (80%) tested positive for the presence of *Coxiella* DNA in aerosols, dust or both (Appendix A). Positive aerosols were detected in 5 farms (1/7 dairy cattle, 2/5 goat herds, and 2/11 mixed herds). The risk of detecting *C. burnetii* DNA in aerosols was associated with recent abortions (estimate 2.485; z value 1.937; *p*= 0.05275; OR = 12.00). Concerning the dust collected from surfaces in the animal premises, *C. burnetii* DNA was detected in 6/7 cattle herds, 2/5 goat herds, 2/2 sheep flocks and 6/11 mixed herds. The risk of detecting *C. burnetii* in dust was significantly associated with the productive aptitude of the herds, with dairy herds showing higher risk compared to meat-producing herds (estimate 1.132; z value −2.369; *p* = 0.0178; OR = 14.7). The remaining variables included in the models did not show a significant association.

### 3.3. Seroprevalence in Wildlife

Twenty-one of the 327 wild ungulates showed antibodies against *C. burnetii* (Table 3 and Appendix A). The highest seroprevalence was observed in red deer (8.43%, 95% CI: 3–14) and the lowest in roe deer (3.51%, 95% CI: 1–8). Regarding the risk of exposure to *C. burnetii*, no significant associations were observed for any of the variables included in the model.

### 3.4. Estimation of Q Fever Incidence in Humans

A total of 1312 patients from Health Centers and hospital admissions showing symptoms compatible with Q fever were included in the study (Appendix A). Of them, 226 were CLIA-positive (17.2%, 226/1312), but only 208 (144 men and 64 women) had IFA titers ≥ 1/128 (Table 4) and were, therefore, considered probable acute Q fever cases.

The number of cases mainly concentrated among the age groups 41–60 years (30.3%) and 61–80 years (40.9%). The distribution of cases was constant throughout the year, with 35% of cases occurring in March–May, 32.7% in June–October and 31.7% in November–February (Table 4). Geographically, the incidence was higher in western HA (54.2 cases/100,000 inhabitants) compared to central (29.3 cases/100,000 inhabitants) and eastern HA (32.6 cases/100,000 inhabitants).

Figure 1 shows a hierarchical representation of the analyzed variables showing an association between age and geographical area. 

MCA analysis identified the most important associations among the categorical variables. A graphic presentation constructed in a series of 2-dimensional spaces is shown in Figure 2.

The two first principal factors derived from the MCA analysis were retained to plot the coordinates of the studied variables and categories. Factorial axis1 (dimension 1) captured 16.1% of the variability and showed a geographical location gradient (east-central-western Asturias). The second axis (dimension 2) captured 15.3% of the variability and showed a gradient in the seasonal appearance of cases. The two dimensions 1 and 2 are sufficient to retain 32.0% of the total inertia (variation) contained in the data. As shown in Figure 2, Q fever cases among older patients in the western region mainly occurred in summer, cases among 60–80-year-old patients concentrated in winter, and 41–60-year-old cases in the eastern region were associated with spring. Q fever cases in younger people (<40 years old) were mainly found in the central region during the summer and spring months. No associations were found with patients’ gender.

Incidence of probable human Q fever cases and seroprevalence against *C. burnetii* in domestic ruminants per geographical region are compiled in Figure 3.

## 4. Discussion

The incidence of human zoonotic infections, like Q fever, reflects the circulation of the bacteria in the animal reservoirs, i.e., domestic ruminants and several wildlife species [1,5]. Therefore, Q fever prevention strategies should incorporate professionals from human health, animal health, and environmental health integrated into a “One Health” approach [24]. In the study area, representing the Cantabrian coast regions, livestock production and hunting activities related to wild ungulates are very important, meeting the conditions for studying *C. burnetii* infection at the wildlife–livestock–human interface.

Contact with domestic ruminants is considered one of the most relevant risk factors in human *C. burnetii* infections [3]. To the best of our knowledge, no previous data on the status of Q fever in domestic ruminants was available in this area, except for a study conducted almost 20 years ago in sheep that showed 5.6% seroprevalence using the complement fixation test (CFT) [25]. ELISA, the technique currently used in most seroprevalence studies, is much more sensitive than CFT [26]. The results obtained in the current study showed that, in general, seroprevalence in domestic ruminants is higher compared to that observed in wild ungulates, suggesting that livestock, and particularly goats, may be the most important reservoir of infection in Asturias. In fact, the most important Q fever outbreak reported in Europe, which occurred in the Netherlands, was associated with goats [2,4], as were the most recent outbreaks reported in the Basque Country, a nearby region in northern Spain [27,28,29].

The seroprevalence values detected in sheep were similar to those observed in this species in other areas of the Iberian Peninsula (11.4%) [30] and comparable to those described in other regions in northern Spain (8.44% vs. 11.8%) [31]. Other Spanish areas, such as the Canary Islands, where the incidence of Q fever in humans is high [7,8], have shown higher seroprevalence in goats and sheep (60.4% and 31.7%, respectively) [32] compared with the studied area. Interestingly, the risk factors associated with increased exposure to *Coxiella* in sheep were the size of the herd and the year of sampling, with significantly higher prevalence in 2018 compared to previous years. In goats, the model found an association with geographical location. Although the risk of transmission of *C. burnetii* from small ruminants to humans seems to be higher than from cattle [3], the current study highlights that the role of cattle as *Coxiella* reservoir must not be underestimated.

Wildlife species are of paramount importance in Asturias due to their diversity, abundance, and interaction with domestic fauna. A large number of samples and species were analyzed in the current study, and the results indicated a different degree of exposure to *C. burnetii* infection among ungulates, with higher seroprevalence in red deer (8.4%) compared to other wild ungulates. Interestingly, seroprevalence in red deer was similar to that observed in previous studies [13] in the same region. Considering that red deer are widely distributed throughout the whole territory of Asturias, its impact as the reservoir of *C. burnetii* would be higher compared to other species with lower seroprevalences that occupied more restricted areas, like fallow deer (eastern Asturias) or roe deer (western Asturias). The rates of exposure of wild ungulates to *C. burnetii* determined here by ELISA were very similar to the infection rates obtained by PCR in roe deer (5.1%) or wild boar (4.3%) in neighboring regions [33].

Considering the results of seroprevalence, it seems that infection is more active within the domestic cycle than in the wild cycle. The fact that *C. burnetii* DNA was detected in 20 of the 25 farms where environmental samples were investigated demonstrates the importance of infection within the domestic cycle. Thus, 80% of the farms harbored *Coxiella* DNA in aerosols and/or environmental dust, indicating that a high percentage of ruminant herds may have had an active infection by *C. burnetii* (aerosol positive) at sampling [28], or have suffered it recently (dust positive) [34]. The recent occurrence of abortions is a risk factor for the presence of *C. burnetii* in aerosols, as has been observed in other studies [35]. The production system also appears to be a key factor for *C. burnetii* infection, with a higher risk in milking herds than meat herds. More intensive management systems where animals remain indoors favor contact between the animals, thus increasing the transmission of the bacteria, especially at lambing/calving time [36].

Considering the high prevalence observed in domestic ruminant farms and the challenging natural environment that facilitates contact with wild species, a high degree of exposure to *Coxiella* would be expected in the local population. This was confirmed by the IFAT results that showed that 15.9% (208/1312) of the people attending health centers with suspected symptoms of Q fever had IFAT titers ≥128 suggestive of probable acute infection. Unfortunately, no paired sera samples were collected 2–3 weeks apart to assess seroconversion, and therefore, it was not possible to confirm diagnosing Q fever [37]. A total of 208 probable cases of Q fever were detected along the study year (2018), representing 33.3 cases/100,000 inhabitants.

Age, geographical location and season were associated with Q fever exposure. In the eastern region, where ovine census is the largest and goats are also present at high numbers, the prevalence among 41–60-year-old patients was higher in spring. The abundance of small ruminants (the main reservoir of *C. burnetii*) in the area may pose a high risk of infection for humans [2,3,4]. Moreover, human Q fever cases in other Spanish regions have been associated with the months following the peak of the ovine lambing [38], which in general concentrated in spring. In the western region of Asturias, the number of farms is smaller, but herd size is larger, cattle predominates over small ruminants, and management is, in general, more professional [17]. In case of infection, many animals (herds with a large census) favors environmental contamination [39,40] and the risk of spread and transmission of *C. burnetii* to surrounding areas through the wind [41]. Q fever exposition in western Asturias occurred predominantly in summer in older patients (81–100 years old) and in winter in 61–80-year-old patients. The wider seasonal distribution might be because calvings occur all along the year in cattle herds. The association of probable cases of Q fever in the elderly with summer is difficult to explain, although summer months might be when older people spend more time in contact with nature. On the other hand, Q fever cases in younger people (<40 years old) were mainly found in the central region during the summer and spring months. Central Asturias is mostly urban, and summer and spring are the seasons when the population spends more leisure time outdoors in nature, thus increasing the possibility of exposure to rural infection sources. In addition, the incidence of Q fever is generally lower under the age of 30 years [7,42], in agreement with the results reported here. Similarly, the incidence is higher in men compared with women [1,7]. This was also the case in this study, though gender was not identified as an explanatory variable.

## 5. Conclusions

This work integrated the collaboration of groups working in animal and public health to obtain a global perspective of the Q fever situation in Asturias (northwestern Spain) in domestic ruminants, wildlife and the human population. Serological and molecular techniques provided an estimation of recent exposure to *Coxiella* and identified a wide distribution of *C. burnetii* infection among domestic ruminants. The results presented might help local authorities to set priorities when implementing control measures.

## Figures and Tables

**Figure 1 animals-11-01395-f001:**
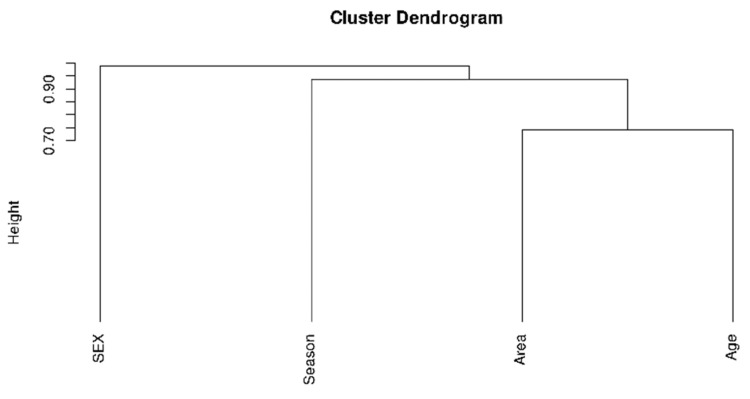
Dendrogram showing the relationship among variables affecting probable cases of Q fever.

**Figure 2 animals-11-01395-f002:**
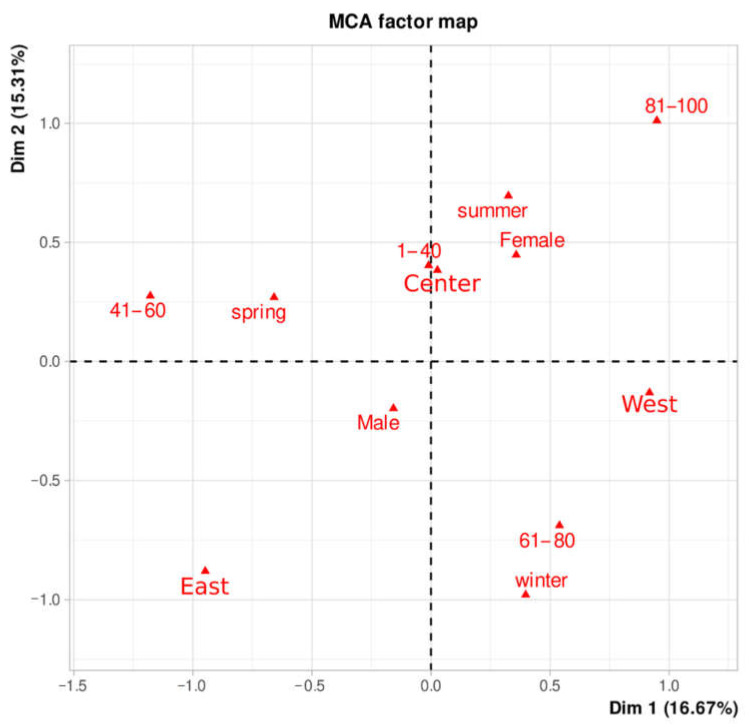
Multiple correspondence analysis describing associations between categories of age, gender, the month of sampling and geographical location of patients with Q fever.

**Figure 3 animals-11-01395-f003:**
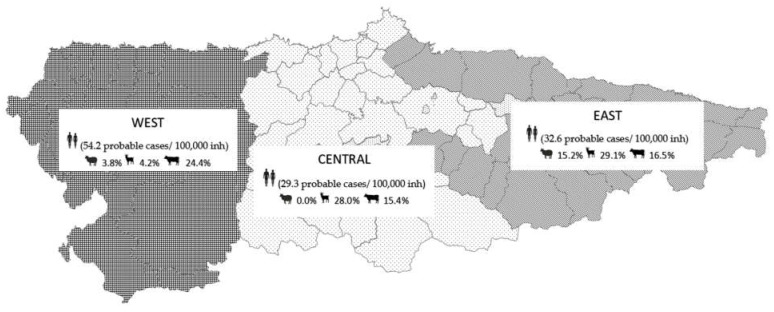
Map of Asturias indicating the number of probable cases of Q fever/100,000 inhabitants and the mean seroprevalence against *C. burnetii* in domestic ruminants in each of the three geographical zones.

**Table 1 animals-11-01395-t001:** *C. burnetii* seroprevalence in domestic ruminants in the three geographical areas of Asturias.

AREA	Sheep				Goats				Cattle			
	Census	Analyzed	ELISA +	*Seropre-*	Census	Analyzed	ELISA +	*Seropre-*	Census	Analyzed	ELISA +	*Seropre-*
	(*n*)	(*n*)	(*n*)	*valence*	(*n*)	(*n*)	(*n*)	*valence*	(*n*)	(*n*)	(*n*)	*valence*
West	7493	26	1	3.8	6609	24	1	4.2	150,250	45	11	24.4
Central	14,557	49	0	0.0	5676	25	7	28.0	132,841	39	6	15.4
East	23,954	79	12	15.2	18,738	86	25	29.1	109,698	79	13	16.5
Asturias	46,004	154	13	8.4	31,023	135	33	24.4	392,789	163	30	18.4

**Table 2 animals-11-01395-t002:** Logistic regression models for the seroprevalence against *C. burnetii* in sheep (A) and goats (B).

**A—Sheep**	**Estimate**	**Z-value**	**Pr (>│t│)**	**OR**	**CI 95%**
Intercept	−3.9802	−4.978	0.0001	0.02	0.01–0.07
Sampling 2016 (ref.)					
Sampling 2017	1.0638	1.260	0.2060	2.90	0.64–20.60
Sampling 2018	2.0249	2.121	0.0339	7.58	1.25–62.66
Census 1–49 animals (ref.)					
Census 50–120	1.9716	3.139	0.0017	7.18	2.10–25.78
**B—Goats**	**Estimate**	**Z-value**	**Pr (>│t│)**	**OR**	**CI 95%**
Intercept	−3.0910	−3.0236	0.0025	0.05	0.01–0.22
Western Asturias (ref.)					
Central Asturias	1.9014	1.7135	0.0866	6.70	1.07–130.56
Eastern Asturias	2.2669	2.1587	0.0309	9.65	1.86–177.50

**Table 3 animals-11-01395-t003:** *C. burnetii* seroprevalence in wild ungulates from Asturias.

Wildlife Species	Analyzed (*n*)	ELISA Positive (*n*)	Seropre- Valence
Iberian red deer (*Cervus elaphus hispanicus*)	83	7	8.43
Roe deer (*Capreolus capreolus*)	57	2	3.51
Cantabrian chamois (*Rupicapra rupicapra*)	41	3	7.32
Fallow deer (*Dama dama*)	73	5	6.85
European wild boar (*Sus scrofa*)	73	4	5.48
Total	327	21	6.42

**Table 4 animals-11-01395-t004:** Distribution of human cases considered as probable Q fever by age and gender, season, and IFAT titer.

AGE	Men	Women	*n*	SEASON	*n*	IFAT Titer	*n*
1–40	17	13	30	Spring (Mar–May)	74	1:128	76
41–60	46	17	63	Summer (Jun–Oct)	68	1:256	56
61–80	62	23	85	Autumn–Winter	66	1:512	44
81–100	19	11	30	(Nov–Feb)		1:1024	31
						1:4096	1
Total	144	64	208	Total	208	Total	208

## Data Availability

Data supporting results can be found in Appendix A.

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
