# Peer review of "One Health Approach: An Overview of Q Fever in Livestock, Wildlife and Humans in Asturias (Northwestern Spain)"

_animals, 2021, doi:10.3390/ani11051395_

Round 1

Reviewer 1 Report

AUTHORS

Manuscript ID: animals-1211318

Title: One Health approach: An overview of Q fever in livestock, wildlife and humans in northwestern Spain

Authors aimed at investigating the seroprevalence of C. burnetii in domestic ruminants, wild ungulates, as well as the current situation of Q fever in humans in northwestern Spain. Authors have found evidence for C. burnetii circulation in animals, humans and environment, suggesting vast circulation. The manuscript is interesting, hence it can potentially be publishable and in that case will provide alerts for public health. The potential impact on infectious diseases control is also high. I have minor comments below

The title suggests that the whole northwestern region of spain was focused however this was only in a small region. I would insert the region before mentioning, northwest Spain in the title

The summary/abstract lacks pointing the aims towards environmental analysis and would gain from it (as it is only focusing on animals and humans)

Do you have an average age for the animals? All were females? Can you better describe the populations?

Line 112: “This required 139 bovine samples, 138 for sheep and 138 112 for goats”. How were these animals chosen within the region (north, west, east and south?) how many per sub-region? How many per farm?

Line 238, please rephrase as it seems mistaken

Figure 3 needs to be horizontal and animal symbols overlap values

Authors compare the distribution of disease between north and south of spain (and also Canary islands) but I think this comparison would gain from including further studies from the whole Iberian peninsula. A quick review of the literature shows many studies in Portugal that could improve the discussion (instead of comparing to canary islands…)

Author Response

Reviewer 1

Manuscript ID: animals-1211318

Title: One Health approach: An overview of Q fever in livestock, wildlife and humans in northwestern Spain

Authors aimed at investigating the seroprevalence of C. burnetii in domestic ruminants, wild ungulates, as well as the current situation of Q fever in humans in northwestern Spain. Authors have found evidence for C. burnetii circulation in animals, humans and environment, suggesting vast circulation. The manuscript is interesting, hence it can potentially be publishable and in that case will provide alerts for public health. The potential impact on infectious diseases control is also high. I have minor comments below

AU: Thank you for the comments. 

The title suggests that the whole northwestern region of spain was focused however this was only in a small region. I would insert the region before mentioning, northwest Spain in the title

AU: Thank you for your comment. In the new version of the manuscript we have included in the title the name of the region where the study was conducted [One Health Approach: An Overview of Q Fever in Livestock, Wildlife and Humans in Asturias (Northwestern Spain)].

The summary/abstract lacks pointing the aims towards environmental analysis and would gain from it (as it is only focusing on animals and humans)

  AU: Due to the limits of the Abstract (200 words), we have added just a few words regarding environmental sampling (line 38-40).

Do you have an average age for the animals? All were females? Can you better describe the populations?

 AU: A systematic random selection of the sera collected in the Livestock Official Sanitary Campaigns was carried out. Unfortunately age information was not recorded, but we know that cattle sera belonged to animals older than 12 months, and older than 6 months in the case of small ruminants. All sera belonged to females. In the new version of the manuscript, we have added a sentence with this information (lines 127-128).

Line 112: “This required 139 bovine samples, 138 for sheep and 138 112 for goats”. How were these animals chosen within the region (north, west, east and south?) how many per sub-region? How many per farm?

  AU: The objective of the study was to estimate the individual seroprevalence against C. burnetii in sheep, goats and cattle, not the herd prevalence. Therefore a systematic random selection of sera from the Livestock Official Sanitary Campaigns sera bank was designed, taking into account the average number of animals per farm, the ruminant species and the geographic location. This type of sampling amounted in general, 1-2 sera per farm. Details of the number of serum samples tested from each animal species in each geographic area are given in Table 1 and Table S1 [Western Asturias = 26 sheep, 24 goats and 45 cattle; Central = 49 sheep, 25 goats and 39 cattle; and East = 79 sheep, 86 goats and 79 cattle].

Line 238, please rephrase as it seems mistaken

  AU: Yes, the reviewer is right. Thank you. The sentence has been modified (line 248-249)

Figure 3 needs to be horizontal and animal symbols overlap values

  AU: The figure 3 has been improved, avoiding overlaps between symbols and values, and placed horizontal in the text.

Authors compare the distribution of disease between north and south of spain (and also Canary islands) but I think this comparison would gain from including further studies from the whole Iberian peninsula. A quick review of the literature shows many studies in Portugal that could improve the discussion (instead of comparing to canary islands…)

 AU: We thought it would be interesting to compare the results of this study with other areas of southern Spain. Data from the Canary Islands were used for comparison because this area has the highest incidence of reported cases of Q fever in humans in Spain and shows very high seroprevalence values in sheep and goats. But the reviewer is right, and comparison of the seroprevalence results with other areas of the Iberian Peninsula is also interesting. In the new version of the manuscript we have discussed the results of seroprevalence in sheep with those obtained by Cruz et al. (2018) in Portugal (line 346). Other studies carried out in Portugal refer to herd seroprevalence (Anastacio et al., 2016; Cruz et al., 2018 TED), thus they cannot be compared with the results of the current study.

Reviewer 2 Report

Lines 85-87 - "The aim of this study was to build a multidisciplinary team" - would be suggested to rephrase the sentence, as the aim of the study should be not to build a team, but to "investigate, analyze, examine or etc."

Lines 235-236 -  "C. burnetii seroprevalence was 8.4% (95% CI: 5-14) in sheep, 18.4% (95% CI: 13-25) 235 in cattle, and 24.4% (95% CI: 18-32) in goats (Table 1)." - from the sentence is not clear if the seroprevalence was calculated based on the all samples collected in 2016-2018 in the three regions analyzed, or the only particular year was included in to analysis.  Additional explanation would be necessary to explain the results presented, while in the Table 2 are presented analysis of the seropositivity in separate years.

Line 245 - "...with seropositivity in sheep (Table 2).", however, the Table 2. presents the seroprevalence against C. burnetii in sheep  and goats. Needs to unify.

Line 255 - "Twenty of the 25 farms (80%) tested positive...." - in the  Materials and Methods was described, that the farms were visited once between February and October 2019 and dust samples were taken from different surfaces of animal premises. Positive aerosols were detected in 5 farms, but from the results presented it is not clear, when the samples were collected, does the season (Winter, Spring, Summer or Autumn) has any influence on the dust collection, was it during the newborn season, or during the season when the most of the abortions occurred. All thus factors might influence on the results, so it is suggested to add some additional explanation to the point 3.2.

Lines 273 - 274 - "A total of 1,312 patients from Health Centers and hospital admissions showing symptoms compatible with Q fever were included in the study." - from the presented data is not clear, does all thus patients have had contacts with domestic ruminants, because in the Conclusion part it is stated, that "Contact with domestic ruminants is considered one of the most relevant risk factors in human C. burnetii infections..."

Lines 279-281 - are the data presented statistically reliable or the results are just mathematically calculated? It should be clearly presented in the results.

Author Response

Reviewer 2

Lines 85-87 - "The aim of this study was to build a multidisciplinary team" - would be suggested to rephrase the sentence, as the aim of the study should be not to build a team, but to "investigate, analyze, examine or etc."

 AU: Thank you for pointing out the mistake. Indeed, the objective was to investigate the situation of C. burnetii from a One health perspective, not to constitute a working team. In the new version of the manuscript, the sentence (lines 86-90) has been corrected.

Lines 235-236 - "C. burnetii seroprevalence was 8.4% (95% CI: 5-14) in sheep, 18.4% (95% CI: 13-25) 235 in cattle, and 24.4% (95% CI: 18-32) in goats (Table 1)." - from the sentence is not clear if the seroprevalence was calculated based on the all samples collected in 2016-2018 in the three regions analyzed, or the only particular year was included in to analysis.  Additional explanation would be necessary to explain the results presented, while in the Table 2 are presented analysis of the seropositivity in separate years.

 AU: Yes, the text refers to the seroprevalence obtained for the total number of serum samples collected for each species, as appears summarized in Table 1. Because a larger number of cows are sampled annually in the sanitary campaigns, the required number of bovine sera was reached with the samplings performed in 2018, but in the case of sheep and goats, due to the smaller census, sera collected in several years had to be included. Consequently, in the logistic regression models, the different categories (years) within the variable “year of sampling” were included in the case of sheep and goats, but not for cattle. The sampling periods for each species are now detailed in M&M (lines 120-128).

Line 245 - "...with seropositivity in sheep (Table 2).", however, the Table 2. presents the seroprevalence against C. burnetii in sheep and goats. Needs to unify.

 AU: Although the same variables were included, the logistic regression models were performed separately for each species (sheep/ goats) and different results were obtained. Table 2 joins the two logistic regression models, one for sheep (A) in which two variables showed significance (sampling year and census), while in the model for goats (B) only the geographic location of the flocks showed statistical differences. Consequently, they cannot be unified. In order to avoid too many tables, we decided to put the results of the two models in the same table, differentiating A and B, as explained in the heading of Table 2.

Line 255 - "Twenty of the 25 farms (80%) tested positive...." - in the Materials and Methods was described, that the farms were visited once between February and October 2019 and dust samples were taken from different surfaces of animal premises. Positive aerosols were detected in 5 farms, but from the results presented it is not clear, when the samples were collected, does the season (Winter, Spring, Summer or Autumn) has any influence on the dust collection, was it during the newborn season, or during the season when the most of the abortions occurred. All thus factors might influence on the results, so it is suggested to add some additional explanation to the point 3.2.

 AU: The database of the environmental study is detailed in Table S1, with the variables included in the statistical analyses. By mistake, in the description of M&M (statistical analyses), it was not mentioned that the variable ‘sampling month’ was also included in the models. Now, this information has been included (line 206-207). In the results section (line 277) it is mentioned that there were no significant association between C. burnetii DNA detection and the remaining variables included in the model.

Lines 273 - 274 - "A total of 1,312 patients from Health Centers and hospital admissions showing symptoms compatible with Q fever were included in the study." - from the presented data is not clear, does all thus patients have had contacts with domestic ruminants, because in the Conclusion part it is stated, that "Contact with domestic ruminants is considered one of the most relevant risk factors in human C. burnetii infections..."

 AU: Unfortunately we do not have this type of information from the patients included in the study. Some of the patients lived in rural areas (western and Eastern Asturias), so the proximity to livestock is likely because Asturias is a region with a livestock tradition. In addition, the abundance of wildlife species and the common practice of outdoor recreational activities in natural parks (i.e., Picos de Europa), favour contact of Asturian population with the natural environment. The literature cites the contact with domestic ruminants as the main risk factor for C. burnetii infection. However, the ability of Coxiella to move through the wind can also give rise to cases and outbreaks in people far from livestock areas.

Lines 279-281 - are the data presented statistically reliable or the results are just mathematically calculated? It should be clearly presented in the results.

  AU: Sorry for this misunderstanding. The results shown in Table 4 and in the text (lines 292-295) are just a mathematical calculation of the distribution of the IFAT positive cases regarding age group, season and IFAT titer. Since negative control cases were not available, no statistical analyses could be done. Thus, we cannot use the word ‘higher’, and it has been replaced by ‘the number of cases mainly concentrated...’ (line 292). Statistical analyses performed on data compiled from the positive IFAT patients have just helped to draw a pattern of the potential risks associated to positive patients.

Reviewer 3 Report

It is  interesting manuscript joining the study livestock, wild animals and environmental samples. The quality of manuscript is very nice to readers. I found small mistake in abstract there is information about 326  sera  from wild ungulates while in the manuscript is information about 327 samples.

Author Response

Reviewer 3

It is interesting manuscript joining the study livestock, wild animals and environmental samples. The quality of manuscript is very nice to readers. I found small mistake in abstract there is information about 326 sera from wild ungulates while in the manuscript is information about 327 samples.

 AU: Thank you, it is an error. Now the abstract has been changed (327 samples).